# Comparative Chloroplast Genome Analysis of Wax Gourd *(Benincasa hispida*) with Three Benincaseae Species, Revealing Evolutionary Dynamic Patterns and Phylogenetic Implications

**DOI:** 10.3390/genes13030461

**Published:** 2022-03-04

**Authors:** Weicai Song, Zimeng Chen, Li He, Qi Feng, Hongrui Zhang, Guilin Du, Chao Shi, Shuo Wang

**Affiliations:** 1College of Marine Science and Biological Engineering, Qingdao University of Science and Technology, Qingdao 266042, China; weicai1123@163.com (W.S.); chenzimeng1209@163.com (Z.C.); qi_feng107@163.com (Q.F.); zhanghongrui1785@163.com (H.Z.); shuowang@qust.edu.cn (S.W.); 2Aesthetic Education Center, Qingdao University of Science and Technology, Qingdao 266042, China; heli2612@163.com; 3Lab of Biorefinery, Shanghai Advanced Research Institute, Chinese Academy of Sciences, Shanghai 201210, China; dugl1@shanghaitech.edu.cn; 4Plant Germplasm and Genomics Center, Germplasm Bank of Wild Species in Southwest China, Kunming Institute of Botany, The Chinese Academy of Sciences, Kunming 650204, China

**Keywords:** *Benincasa hispida*, chloroplast genome, comparative analysis, divergence region, phylogenetic

## Abstract

*Benincasa hispida* (wax gourd) is an important Cucurbitaceae crop, with enormous economic and medicinal importance. Here, we report the de novo assembly and annotation of the complete chloroplast genome of wax gourd with 156,758 bp in total. The quadripartite structure of the chloroplast genome comprises a large single-copy (LSC) region with 86,538 bp and a small single-copy (SSC) region with 18,060 bp, separated by a pair of inverted repeats (IRa and IRb) with 26,080 bp each. Comparison analyses among *B. hispida* and three other species from Benincaseae presented a significant conversion regarding nucleotide content, genome structure, codon usage, synonymous and non-synonymous substitutions, putative RNA editing sites, microsatellites, and oligonucleotide repeats. The LSC and SSC regions were found to be much more varied than the IR regions through a divergent analysis of the species within Benincaseae. Notable IR contractions and expansions were observed, suggesting a difference in genome size, gene duplication and deletion, and the presence of pseudogenes. Intronic gene sequences, such as *trnR-UCU–atpA* and *atpH–atpI*, were observed as highly divergent regions. Two types of phylogenetic analysis based on the complete cp genome and 72 genes suggested sister relationships between *B. hispida* with the *Citrullus*, *Lagenaria*, and *Cucumis*. Variations and consistency with previous studies regarding phylogenetic relationships are discussed. The cp genome of *B. hispida* provides valuable genetic information for the detection of molecular markers, research on taxonomic discrepancies, and the inference of the phylogenetic relationships of Cucurbitaceae.

## 1. Introduction

Cucurbitaceae is a moderately large family of about 130 genera and 900 species [1]. Because of their economic importance in temperate regions, species of the Cucurbitaceae family have a long and close association with human beings [2]. Familiar edible and medicinal fruits, such as cucumber (*Cucumis sativus*), melon (*Cucumis melo*), watermelon (*Citrullus lanatus*), bottle gourd (*Lagenaria siceraria*), pumpkin, and squash (*Cucurbita* spp.) are the main crops of Cucurbitaceae [3]. All are economically valuable fruit crops. The early molecular phylogeny of Cucurbitaceae was reconstructed using five chloroplast markers, which weakly support two subfamilies of Cucurbitoideae and Nhandiroboideae [4]. Recent studies have reported that the phylogenetic tree of Cucurbitaceae contains a new classification of 15 tribes and 95–97 genera, using 14 molecular markers from the nuclear, plastid, and mitochondrial genomes [5]. However, the relationships between these subfamilies are still unresolved, possibly due to limited phylogenetic signals of the molecular markers, with a large proportion (over 70%) of missing data [6]. Comprehensive and complete sequence information is a reliable foundation for phylogenetic studies of Cucurbitaceae.

*Benincasa* represents a monotypic genus with a single species, belonging to tribe Benincaseae (Cucurbitaceae), which is not difficult to find in the markets since: (1) it is a highly commercialized vegetable due to its long shortage properties; and (2) it bears giant fruit, normally 80 cm in length and with a weight of over 20 kg [7,8]. Wax gourds are widely distributed in temperate and sub-temperate climates, such as China, Japan, Korea, India, and several tropical countries. Currently, it is being increasingly popular in the Caribbean and the United States [8]. Wax gourd has important nutritional and medical applications as an important vegetable crop [9]. Its pharmaceutical values cover various aspects, including central nervous system diseases (muscle tension, Alzheimer’s disease [10]), gastroprotective diseases [11], depression-like activities [12], diabetes, dropsy, diseases related to the liver, urinary diseases [13], and heart diseases. Other effects, including hypolipidemic, antioxidant, anti-inflammatory, antipyretic, anti-angiogenic [14] and antimicrobial properties of *B. hispida* are also reported [15,16,17]. Studies have reported that the seeds of *B. hispida* contain saponin, urea, citrulline, oleic acid, and fatty acids [18,19].

The chloroplast (cp) is a self-replicating organelle that consists of homogeneous circular DNA molecules. The double-strand DNA inside the cp genome ranges from 70 to 520 kb in algae and is generally more conserved in land plants, ranging from 120 to 160 kb. Although the specific nucleotide sequences vary across different species, the quadripartite structure and organization retain a firm consistency, which can be classified into four sections: a large single-copy (LSC) region and a small single-copy (SSC) region. These are separated by a pair of inverted repeats (IRa and IRb) [20]. As a metabolic center, the cp genome remains highly conservative to sustain the normal physiological function of cells, especially for genes related to photosynthesis. Despite its conservative nature, there are partial differences in gene types and genome sizes, such as substitutions, insertions, and deletions of nucleotide sites; contractions and expansions of IR regions; and rearrangements and translocations of genes [21,22]. This polymorphism and diversity can be used in population taxonomic and phylogenetic analysis, population genetics studies, and evolutionary investigations [23,24].

Although wax gourd is widely consumed among Asian communities, no detailed resources have been published regarding its genomic features, nor has a comparative analysis with its related species. To date, the nuclear gene of wax gourd has been reported [25], whereas the whole nuclear genome has some limitations in phylogenetic analysis of species. Therefore, we sequenced and assembled the complete chloroplast genome sequence of *B. hispida* and submitted the data to the National Center for Biotechnology Information (NCBI). This study is the first elaborate report of the cp genome of *B. hispida*, and the first comparative analysis to include three other species from Benincaseae, namely *Lagenaria siceraria*, *Citrullus colocynthis*, and *Citrullus lanatus*. We aimed to reveal: (1) the quadruple structure and the composition of different regions and functions; (2) putative RNA editing sites; (3) patterns of repeats and microsatellite; (4) highly divergent regions; (5) the phylogenetic relationships among Cucurbitaceae. The results provided may contribute to the unfolding of taxonomical discrepancies, identifying suitable genetic markers, and the inference of phylogenetic positions among related species.

## 2. Materials and Methods

### 2.1. Plant Material, DNA Extraction, and Sequencing 

Fresh leaves of *Benincasa hispida* were collected from Panlong District, Kunming City, China (24°23′ N, 102°10′ E), and the voucher specimen and DNA were deposited at Qingdao University of Science and Technology (specimen code DG200618). Fresh leaf tissue was collected without apparent disease symptoms and preserved in silica gel. Total genomic DNA was extracted from fresh leaves using modified CTAB [26], the quantity and quality of extracted DNA was assessed by spectrophotometry, and the integrity was evaluated using a 1% (*w*/*v*) agarose gel electrophoresis [27]. An Illumina TruSeq Library Preparation Kit (Illumina, San Diego, CA, USA) was used to prepare approximately 500 bp of paired-end libraries for DNA inserts, according to the manufacturer’s protocol. These libraries were sequenced on the Illumina HiSeq 4000 platform in Novogene (Nanjing, China), generating raw data of 150 bp paired-end reads. About 14.6 Gb high quality, 2 × 150 bp pair-end raw reads were obtained and were used to assemble the complete chloroplast genome of *B. hispida*.

### 2.2. Genome Assembly and Annotations

The raw data were preprocessed using Trimmomatic 0.39 software [28], including removal of Adapter sequences and other sequences introduced in the sequencing, removal of low-quality and over-N-base reads, etc. The quality of newly produced clean short reads was assessed using FASTQC v0.11.9 and MULTIQC software [29,30], and high-quality data with Phred scores averaging above 35 were screened out. According to the reference sequence (*Cucumis melo*), the chloroplast-like (cp) reads were isolated from clean reads by BLAST [31]. Short reads were de novo assembled into long contigs with SOAPdenovo 2.04 [32] by setting kmer values as 35, 44, 71, and 101. Finally, the long-contigs complete sequence expansion and gap filling using Geneious ver. 8.1 [33], forming the complete chloroplast genome. The complete chloroplast genome was further validated and calibrated by using de novo splicing script NOVOplasty 4.2 [34]. GeSeq [35] was used to annotate the de novo assembled genomes, tRNAscan-SE ver 1.21 [36] was applied to detect tRNA genes with default settings, and RNAmmer [37] was used to validate rRNA genes with default settings. As a final check, we compared the results with the reference sequence and manually corrected the erroneous genes by GB2Sequin [38]. The circular map of the genomes was drawn by using Organellar Genome DRAW (OGDRAW) [39]. The newly assembled *B. hispida* chloroplasts genomes were deposited in GenBank, with the accession number MW362306.

### 2.3. Chloroplast Genome Comparison

In order to gain a better understanding of the characteristics of the cp genome of *B. hispida*, we selected three species that are not only closely related to *B. hispida* but also representative of Benincaseae to perform comparative analysis. Sequences of their complete chloroplast genome were downloaded from NCBI database, with the following accession numbers: *Lagenaria siceraria* (MT773628), *Citrullus colocynthis* (NC_035727), and *Citrullus lanatus* (KY430692).

### 2.4. Codon Usage and Putative RNA Editing Site

Codon usage and amino acid frequency were calculated by Geneious Prime^®^ 2020 [40], and relative synonymous codon usage (RSCU) of protein-coding genes was evaluated by MEGA-X [41]. We also used predictive RNA editors for plant chloroplast (PREP-cp) [42] to investigate putative RNA editing sites in the cp genomes of *B. hispida*, *C. colocynthis*, *C. lanatus*, and *L. siceraria*.

### 2.5. Repeat Sequences and SSR Analysis

MIcroSAtellite identification tool (Misa) [43] was used to determine simple sequence repeats (SSRs) or microsatellites in cp genomes of four species. SSRs were determined by a settled minimum threshold of nine for mononucleotide repeats, four for dinucleotide, and three for tri-, tetra-, penta-, and hexanucleotide repeats. Oligonucleotide repeats were analyzed by REPuter program [44] to find four types of repeats, including forward (F), reverse (R), complementary (C), and palindromic (P). These four types of repeats were detected with a minimum repeat size of 20 bp, edit distance of 3, and 90% similarities.

### 2.6. Comparative Analysis of cp Genomes in Benincaseae

IRscope [45] was used to detect the contraction and expansion of IRs boundaries, which were visualized between four main regions in chloroplast genome (LSC/IRb/SSC/IRa). The mVISTA program [46] was used to compare the cp genome of four species using Shuffle-LAGAN model with *Lagenaria siceraria* set as the reference sequence.

DnaSP [46] was used to perform sliding window analysis using multiple alignment of complete cp genome of four selected species, along with the determination of synonymous (Ks) and non-synonymous (Ka) substitutions and their ratio (Ka/Ks). Geneious was used to detect the types, numbers, lengths, and positions of SNPs and InDels in LSC, SSC, and IR regions. To further evaluate their natural selection pressure, genes that presented Ka/Ks value greater than one were tested with site model using CodeML [47] algorithm implemented in EasyCodeML [48]. The likelihood ratio test (LRT) was used to compare seven codon substitution models (M0, M1a, M2a, M3, M7, M8, and M8a). The Bayes empirical Bayes (BEB) evaluated the posterior probability of positive selection sites.

### 2.7. Phylogenetic Analysis 

We selected and downloaded the sequences of 23 species from Cucurbitales and three outgroup species including *Libidibia coriaria* (NC_026677), *Glycine max* (NC_007942) and *Solanum lycopersicum* (NC_007898) from NCBI to perform phylogenetic tree building. Maximum likelihood (ML) tree was constructed through two approaches. One phylogenetic tree was constructed using complete cp genome and the other was built with 72 gene sequences. MAFFT alignment was performed using 72 concatenated gene sequences and the best-fit model was found by MEGA-X [41] All indels was excluded for both alignments, leaving only substitutions for ML analysis. The best-fit models applied for all three were GTR + G, determined based on Bayesian information criterion (BIC) [49].

## 3. Results

### 3.1. Chloroplast Genome Assembly, Organization, and Features of Benincasa Hispida

The paired-end sequencing of *B**. hispida* by Illumina HiSeq 4000 generated around 14.6 GB raw data with 82.6 million 150 bp reads. We de novo assembled its complete chloroplast genome and the data were submitted to NCBI under accession number MW362306 after a thorough check for correctness. As shown in Table 1 and Figure 1, the size of its complete chloroplast genome is 156,758 bp in length, presenting a typical quadripartite structure with a large single-copy region (LSC, 86,538 bp), a small single-copy region (SSC, 18,060 bp) and two inverted repeat regions (IRa/b, 26,080 bp each). 

The cp genome of *B. hispida* had 131 genes (Table 2), including 86 protein-coding genes, 37 tRNA genes, and 8 rRNA genes, 18 of which were duplicated genes (7 protein-coding genes, 7 tRNA genes, and 4 rRNA genes). The total GC content of the cp genome was 37.2%, with the IR regions having the highest GC content at 42.9%, followed by LSC (35%) and SSC (31.7%). In terms of the GC contents of the different gene types, the number of rRNA (55.2%) and tRNA (53.2%) was relatively high, and that of CDS was 37.9%. In total, 18 genes contained introns, 16 of which (10 protein-coding genes and 6 tRNA genes) contained 1 intron, and 2 CDSs (*ycf3* and *clpP1*) possessed 2 introns (Appendix A). Among these genes, 17 genes were duplicated in the IR regions except one trans-splicing gene, which was observed in the *rps12* gene with 5’–end located in the LSC region and 3’ end duplicated in the IR regions. The truncation event was observed in the *ycf1* gene, which started in the IRa region and ended at the SSC region, leaving a 100 bp truncated copy in the IRb region.

### 3.2. Codon Usage and Amino Acid Frequencies

The complete cp genome of *Benincasa hispida* contained 80,109 bp of coding sequences (CDSs) that encoded 86 genes, including 26,703 codons that fit in 64 codon types. The results of the amino acid frequency analysis showed that leucine, with 10.5% occurrence, was the most abundant amino acid, followed by isoleucine, with 8.5%. Cysteine, with only 1.1% abundance, was the amino acid that occurred the least. 

The relative synonymous codon usage (RSCU) of the four species was also calculated, presenting a high codon bias of A or T bases. The distribution of the codon usage showed that the codons ending with A or T had RSCU > 1 except GGT (Glycine, 0.96), AGT (Serine, 0.9), and CGT (Arginine, 0.68), revealing that the codons ending with A or T were preferred, while the codons ending with C or G were non-preferred. Among all three stop codons, TAA, with 64% abundance, was the most frequent (Appendix A).

### 3.3. Putative RNA Editing Site within Benincaseae

RNA editing events are typical in the cp genomes of most land plants and essential for understanding the chloroplast genome at the transcript level. For this purpose, we determined the RNA editing site in the cp genomes of four species from Benincaseae. In the cp genome of *Benincasa hispida*, PREP-web found 58 putative RNA editing sites in 21 CDS (Appendix A). Among these genes, the *ndhB* gene, with thirteen editing sites, was determined to be the most variant gene, followed by *ndhD* (eight sites) and *rpoB* (five sites). We also found that 81% of all RNA editing events occurred at the second nucleotide position of the codons, while none of these events were located in the third codon position. 

Moreover, these RNA editing events resulted in post-transcriptional substitutions, causing amino acid conversions. In the group of these conversions, fifty-four out of fifty-six RNA editing sites led to hydrophobic products, comprising phenylalanine (9), isoleucine (5), leucine (32), methionine (2), valine (4), and tryptophan (2). Four exceptions led to hydrophilic (neutral) amino acid products, including cysteine (1), tyrosine (2), and serine (1). Furthermore, serine-to-leucine was found to be the most abundant post-transcriptional substitution, with 41.82% of all RNA editing events, followed by proline-to-leucine (14.55%) and serine-to-phenylalanine (7.27%). It is worth mentioning that two RNA editing events were detected that transformed ACG (Thr) to the initiation codon AUG, resulting in the start of translation in the *ndhB* and *ndhD* genes.

As shown in Appendix A, the total number of RNA editing sites detected was 57 in *Citrullus lanatus* and 55 in *Lagenaria siceraria* and *Citrullus colocynthis*. All the patterns mentioned above showed high consistency in all four species analyzed, with only minor differences in terms of numerical values.

### 3.4. Repeated Sequence and SSR Analysis

In this study, we analyzed microsatellites or simple sequence repeats (SSRs) in the cp genome of *Benincasa hispida*, *Citrullus lanatus*, *Lagenaria siceraria*, and *Citrullus colocynthis* using MISA-web, and high similarity was revealed between the four species (Figure 2). We found that *B. hispida* contained the most abundant number of SSRs (238), while *C. lanatus*, with only 219 SSRs, had the least. In the cp genome of *B. hispida*, most of the SSRs were mononucleotides (42%), varying from 9 to 15 repeat units. Meanwhile, the abundance of dinucleotide was only 25%, which was slightly lower than that of trinucleotide (30%). The frequencies of tetranucleotide and pentanucleotide were only 3% and 0.42%, respectively, and hexanucleotide repeats were absent from all the species (Figure 2C). Moreover, most of the mononucleotide repeats were A/T motifs, while AT/TA motifs comprised 68% of the dinucleotide repeats (Appendix A).

We also analyzed the distribution of SSRs in two different types of regions, specifically LSC/IR/SSC regions and intergenic spacer (IGS)/gene regions. According to the results, most of the repeats were located in the LSC region, varying from 136 in *C. lanatus* to 148 in *B. hispida*, followed by the SSC region (38 in *B. hispida*) and IR regions. Noticeably, the SSC number in the IR regions was 26 in all the species except for *L. siceraria* (24), implying that the IR regions were more conserved than the LSC and SSC regions (Figure 2A). The IGS regions were determined to have a high abundance of SSRs in comparison with the gene regions. We found 125 SSRs within 46,150 bp IGS regions and 116 SSRs in 112,281 bp gene regions, meaning the density of SSRs in the IGS regions was 2.62 times of that of the gene regions (Figure 2B). Similar results were present in all the species.

The oligonucleotide repeat sequences were also analyzed using the REPuter program to detect the abundance of four types of oligonucleotide repeat, including forward (F), palindromic (P), reverse (R), and complementary (C). Although minor variations presented about the total number of oligonucleotide repeats, the distribution of the four types of repeats and the size of the repeats presented an obvious resemblance. In terms of the number of oligonucleotide repeats and their distribution in each type, we found 42 repeats (F = 16, P = 22, R = 4) in the cp genome of *B. hispida*; 41 (F = 16, P = 21, R = 4) in *C. lanatus*; 46 (F = 14, P = 26, R = 4) in *L. siceraria*; and 42 (F = 14, P = 23, R = 5) in *C. colocynthis* (Figure 2A). The length of repeats was mostly found between 20 and 24 bp (Figure 2C). The palindromic repeats were the most abundant repeats, followed by forward repeats, whereas the number of reverse repeats was low. None of the species had complementary repeats. We also located the region of each oligonucleotide repeat; significant consistency was presented among the four species. The number was exactly the same in all the species regarding the repeats located in the IR regions (6) and some shared sequences, including sequences between LSC and IRa/b (4), between SSC and IRa/b (2), and from IRb to IRa crossing SSC (Figure 2B). 

### 3.5. IR Contraction and Expansion 

The genome length of the chloroplast ranged from 159,758 bp (*B. hispida*) to 157,147 bp (*C. colocynthis*). Furthermore, in the cp genome of *B. hispida*, the length of the IR regions was the shortest with 260,080 bp, while that of the SSC region was the longest with 180,060 bp (Appendix A). Thus, we inferred that the variation in size of the cp genome was contributed to by the expansion and contraction of the IR regions (Figure 3). The junction sites between each region were denoted as: J_LB_ (IRb/LSC), J_SA_ (SSC/IRa), J_SB_ (IRb/SSC), and J_LA_ (IRa/LSC). All eight species analyzed presented functional *ycf1* genes, six of which were at J_SA_, while the other two were located in the SSC region completely. Moreover, the *ycf1Ψ* (pseudo-copy) was only present in two species (*B. hispida* and *L. siceraria*) at J_SB_ and were 3 bp and 25 bp in the SSC region, respectively. The *ndhF* gene was revealed in all species in J_SB_ with the same length (2246 bp) and relatively consistent position with only a few bp in the IRb region, except *C. hystrix*, with 21 bp. and *B. hispida* (completely located in the SSC region). 

The *rpl2* gene was found close to J_LB_, while that of two species *(C. moschata* and *C. lanatus*) were in the LSC region with 11 bp and 6 bp, respectively. At the same time, the duplicate *rpl2* genes were absent in the same two specific species. The *rps19* gene was the most variant gene among all the genes close to the IR junction. In the four species, the *rps19* genes were 2 bp in the IRb region and the remaining four were completely in LSC region. 

### 3.6. Divergence Analysis of Chloroplast Genome 

To identify the diversity in the chloroplast genomes of four Benincaseae species, we visualized the percentage of identity between the sequences and colored regions of high conservation using mVISTA program. As shown in Figure 4, the sequences varied remarkably among different regions. Firstly, most of the differences were located in the LSC and SSC regions, while the IR regions were almost identical among the four species except the *rps12* gene, revealing that the IR regions were more conservative than the LSC and SSC regions. Moreover, the IGS regions revealed themselves to be remarkably more divergent than the gene regions. Notable divergent non-coding regions included: *trnR-UCU–atpA*, *atpH–atpI*, *trnT-GGU–psbD*, *trnL-UAA–trnF-GAA*, *accD–pasI*, and *ndhF–rpl32*. Genes such as *ycf1*, *ycf2*, *accD*, *psbA*, *ccsA*, *ndhF*, and *matK* were found to be highly divergent coding genes. 

The Ka/Ks ratio is an essential index to identify a mutation as neutral, purifying, or beneficial. Thus, we compared *B. hispida* with *C. colocynthis*, *C. lanatus*, and *L. siceraria*, respectively, to analyze the synonymous substitutions (Ks), the non-synonymous substitutions (Ka), and their ratio (Ka/Ks) of 73 PCGs (Appendix A). In total, 18 genes could not be determined due to absent information (Ks = 0). After deleting these genes, as well as the non-substitution results, we found that the genes carrying out photosynthesis functions revealed Ka/Ks = 0 or at relatively low values, indicating that these groups of genes were fairly conserved. The Ka/Ks ratio of 26 genes was lower than 0.5 and that of 96% genes was lower than 1, with only 5 exceptions (*accD*, *clpP*, *rps4*, *ycf1*, and *ycf2*). We then performed a purifying/positive selection site evaluation for these five genes (Appendix A). However, only two genes o, *accD* and *rps4*, presented sites potentially under positive selection, indicated by the high empirical Bayes values (Table 3).

To obtain a holistic understanding of the sequence divergence, we performed a sliding window analysis to visualize the nucleotide variability values of all the cp genomes. We found that none of the π values of the CDS genes exceeded 0.05 and that the IGSs were more divergent than the gene regions, which was consistent with the aforementioned analysis. It can be clearly seen in the figure that the SSC and LSC regions were much more divergent than the IR regions, the π value of which was remarkably low and mirror-symmetrized with SSC as the center (Figure 5).

### 3.7. Phylogenetic Analysis

To locate the phylogenetic position of *B. hispida* precisely, we selected 26 species (Appendix A) and constructed two phylogenetic trees using the complete cp genome (Figure 6A) and 73 selected CDSs (Figure 6B), respectively. The results all suggested that *B. hispida* was closely related with *Cucumis*, *Citrullus*, and *Lagenaria* as their sister group, with fairly high bootstrap values. The phylogenetic relationship results of the two approaches presented high consistency, with two main variations. Firstly, in general, the bootstrap values in the tree that applied the complete cp genome were higher than in the tree constructed with 73 CDSs (Figure 6B). In addition, Begoniaceae was a sister group with Coriariaceae and Corynocarpaceae, according to Figure 6A, while in Figure 6B, Coriariaceae and Corynocarpaceae were the early-diverging lineages of Begoniaceae. However, only 82 bootstrap values supported the former situation (Figure 6A) while 94 supported the second (Figure 6B).

## 4. Discussion

In this study, we sequenced and reported the complete chloroplast genome of *Benincasa hispida* and performed comparative analyses with three other, closely related species selected from the *Benincaseae* but distinct enough to obtain reasonable results, providing valuable genetic data for phylogeographic and population genetic investigation [50,51].

The cp genome revealed high consistency in terms of its quadruple structure, gene content, and organization not only in Benincaseae [52,53], but also in other angiosperms [54]. The genome size differed less than 400 bp, with almost identical gene numbers, signifying that the cp genomes among the four analyzed species were conservative on the whole. The GC content of *B. hispida* varied across different regions and functions. The rRNA sequences were considerably rich in GC bases; as a consequence, the IR regions rich in rRNA appear to have had higher GC content than the other regions. These findings agree with those of previous studies [55,56].

However, changes were found that provide valuable information for understanding the development and evolution [50,57]. The bias of the codon usage in the plant cp genome was an important evolutionary feature for the studies regarding mRNA translation, new gene discovery, and molecular biology [58]. Previous studies have confirmed that genes tend to choose preferred synonymous codons for specific amino acids rather than randomly distributions [59,60]. Our study showed that genes of *B. hispida* prefer codons with A/T in the third position, which was consistent with previous studies [61,62].

Microsatellites, or SSRs, are widely distributed in cp genomes that serve as molecular markers for phylogenetic relationship inference [63,64]. Moreover, SSRs are also related to different types of genome rearrangement, recombinations, and large inversions [65,66]. Similar to previous studies, we found that mononucleotide repeats were the most abundant types of repeat and that their numbers in the LSC region far surpassed those in the SSC and IR regions [67]. Furthermore, a greater number of palindromic repeats were found among four types of repeat, while previous studies revealed that the forward repeats were the most abundant repeats [61,68]. We specifically analyzed the abundance of SSRs that differed from gene regions to intronic gene regions and verified that the IGSs contained much higher SSR density than the others. Thus, we inferred that IGS regions may undergo gene rearrangement and recombination more frequently than gene regions. Moreover, our results support the hypothesis that cpSSRs are more often composed by polyA or polyT than polyG or polyC [69,70], implying that IGSs might be relatively rapidly mutating regions [71,72].

It is commonly agreed that variations in genome size in the chloroplast are the consequence of IR contraction and expansion, leading to gene duplication and deletion and the presence of pseudogenes [68,73]. We found that *ycf1Ψ* pseudogenes were only detected in *B. hispida* and *L. siceraria*, which were also sister groups in the ML phylogenetic trees. Furthermore, no *rps19Ψ* pseudogenes were observed in any of the species analyzed; their presence was thought to be responsible for the loss of function of the *rps19* gene [74,75]. These results imply that gene variation at IR boundaries may contribute to the understanding of the cp genome at a molecular level and serve as an indicator for evolutionary investigation [76,77].

It is worthwhile to study the genetic diversity among the four Benincaseae species because the chloroplast genome plays a crucial role in the study of phylogeny, gene flow between species, and population genetics among different species. [64,78]. The coding regions were generally found to be more conserved than the non-coding regions. Furthermore, some of the coding genes, namely the *ycf1*, *ycf2*, *matK*, *accD*, and *ndhF* genes, were commonly found to be relatively divergent [79,80]. In addition, the LSC and SSC regions were further confirmed to be more divergent in comparison with the IR regions [81]. We also discovered that genes related to photosynthesis with low Ka/Ks ratios showed slow evolution rates, while functional genes, such as *accD*, revealed high evolutional rates, indicating that genes carrying out vital functions were conserved and vice versa [74,82]. Among the five genes that showed Ka/Ks values greater than one, two genes, *accD* and *rps4*, presented one positive selection site, respectively. These results indicate that the *accD* gene may have changed under evolutionary pressure [83]

Currently, protein-coding genes are commonly implemented for phylogenetic tree building [84]. While the complete cp genome contains richer information but requires a longer time to perform, higher-end equipment and the population distance may be exaggerated for the highly divergent features of IGS genes [85]. In this study, we applied both methods to build the phylogenetic trees. The first tree, built with the complete cp genome, revealed higher bootstrap values in general, while the other tree, built with coding genes, showed a slightly different phylogenetic order in four species, out of twenty-six in total. In general, the phylogenetic position revealed was consistent with previous studies [86,87,88,89]. However, the phylogenetic relationships we discovered within Cucurbitaceae differed from the results of previous phylogenetic marker-based taxonomy research [90,91]. This may have been due to the different approaches used for phylogenetic tree construction, with further investigation needed.

In conclusion, our study first shed light on the structure and content of the cp genome of *B. hispida*, an economically important fruit crop widely distributed in several tropical countries and extensively consumed worldwide. We also offered information regarding similarities and divergence, enriching the understanding of the species of Benincaseae. Moreover, information about highly polymorphic regions was also provided regarding molecular markers and highly divergent regions, which might be useful for further studies of the taxonomy and phylogeographics of Benincaseae subfamilies.

## Figures and Tables

**Figure 1 genes-13-00461-f001:**
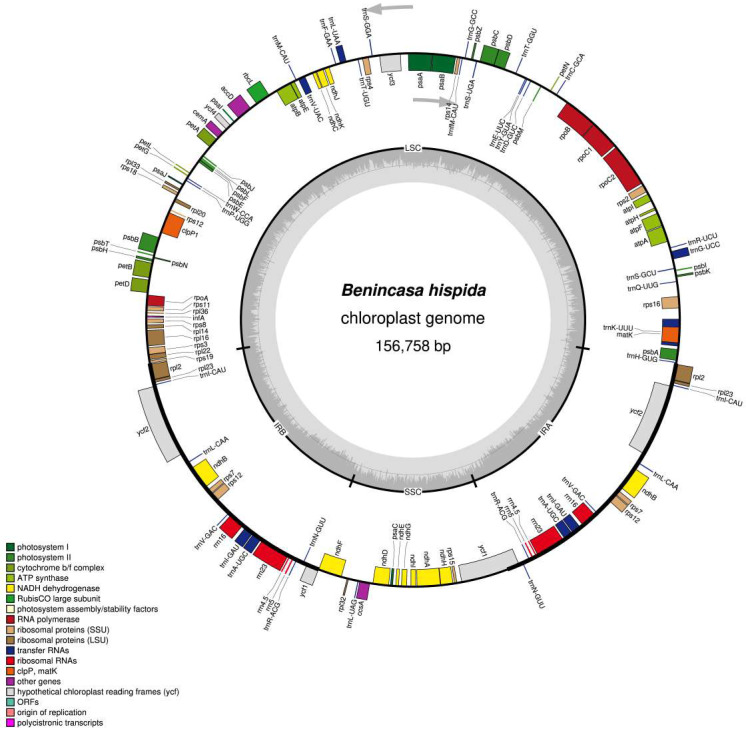
Gene map of the *Benincasa hispida* chloroplast genome. The genes drawn outside and inside of the circle are transcribed in clockwise and counterclockwise directions. Genes are colored based on their function. The borders of chloroplast genome are defined with LSC, SSR, IRa, and IRb. The dashed gray color of the inner circle shows the GC content, whereas the lighter gray color shows AT content. Asterisks mark genes that have introns.

**Figure 2 genes-13-00461-f002:**
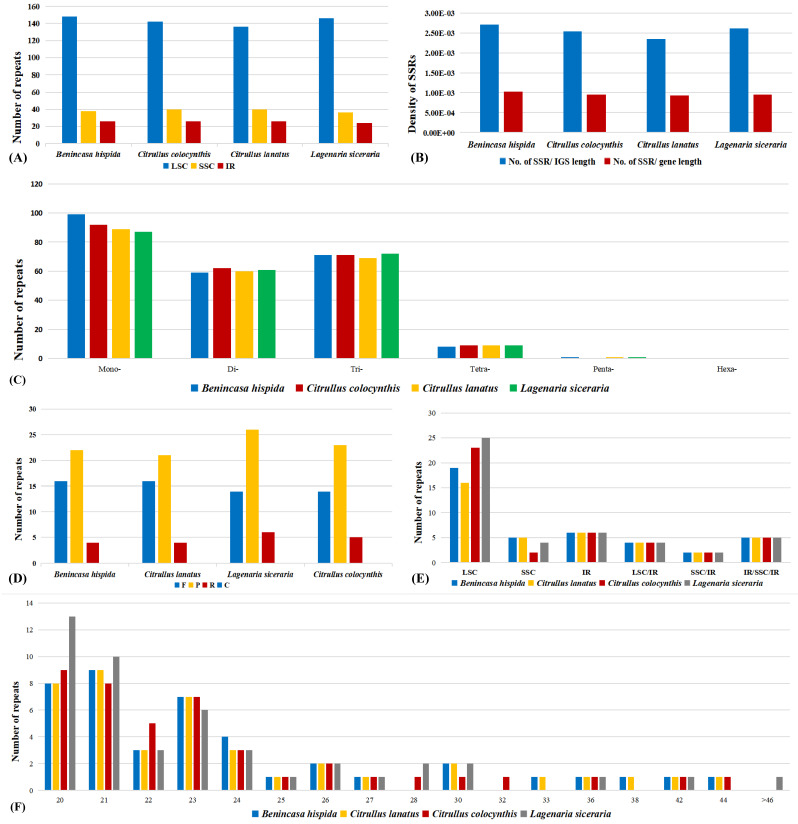
Comparison of microsatellites and oligonucleotide repeats in the chloroplast genomes of *Benincaseae species*. (**A**) The number of SSRs in the three main regions of the chloroplast genome. LSC: large single-copy region, SSC: small single-copy region, IR: inverted repeat region. (**B**) The density of the SSRs in the IGSs (intergenic sequences) and gene regions. (**C**) The number of different types of SSR. Mono- represent mononucleotide SSRs, Di- represent dinucleotide SSRs, etc. (**D**) Different types of oligonucleotide repeat. F: forward repeats, P: palindromic repeats, R: reverse repeats, C: complementary repeats. (**E**) The number of oligonucleotide repeats in different regions. LSC: large single-copy region, SSC: small single-copy region, IR: inverted repeat region, LSC/IR: repeat sequences crossed LSC and IR regions, etc. (**F**) The number of repeats in different repeat units.

**Figure 3 genes-13-00461-f003:**
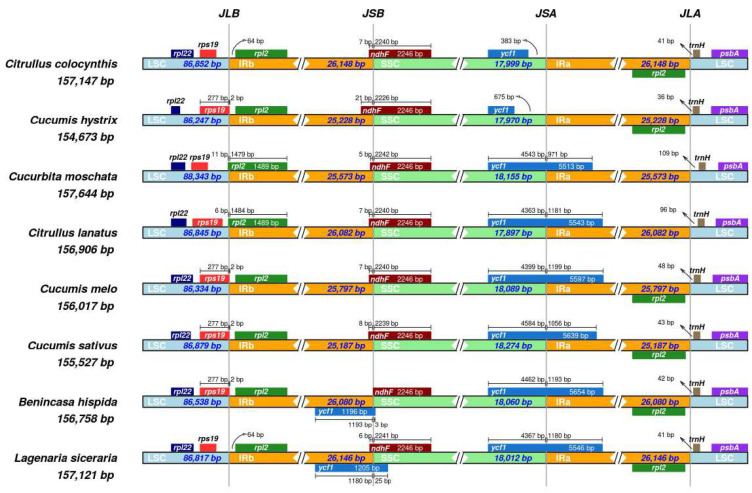
Comparison of junctions between the LSC, SSC, and IRs among eight species. Number above indicates the distance in bp between the ends of the genes and the border sites (distances are not to scale in this figure).

**Figure 4 genes-13-00461-f004:**
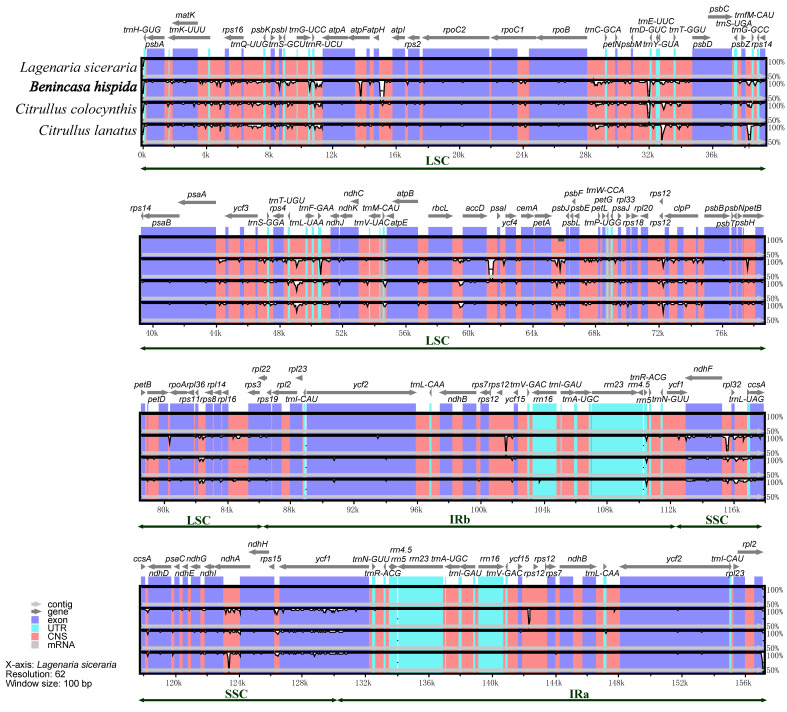
Sequence identity plot comparing the chloroplast genomes among *Benincaseae species* with *Lagenaria siceraria* set as a reference using mVISTA. Pink bars represent noncoding sequences (CNS), and white peaks represent genome divergence. The y-axis represents the percentage identity (shown: 50–100%).

**Figure 5 genes-13-00461-f005:**
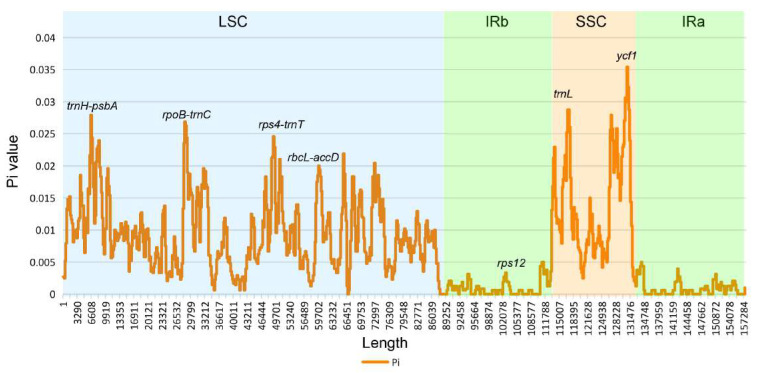
Nucleotide diversity (π) values among the *Benincaseae* species.

**Figure 6 genes-13-00461-f006:**
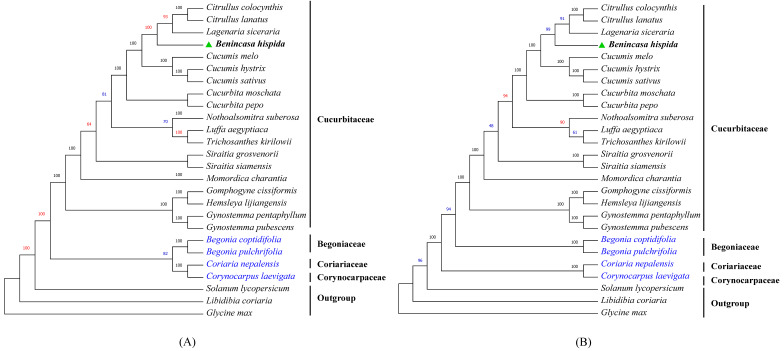
Maximum likelihood (ML) tree of Cucurbitales. (**A**) The phylogenetic tree constructed by complete chloroplast genome of 23 species. (**B**) The phylogenetic tree builds with 72 genes. The positions of *Benincasa hispida* are marked with green triangles. Numbers above branches are bootstrap values, and the bootstrap values higher or lower than those of the other tree are marked as red or blue, respectively. *Glycine max* set as the root in both trees.

**Table 1 genes-13-00461-t001:** Chloroplast genome general features of *Benincasa hispida*.

Characteristics		*Benincasa hispida*
Size (base pair, bp)		156,758
LSC length (bp)		86,538
SSC length (bp)		18,060
IR length (bp)		26,080
Number of genes		131
Number of protein-coding genes		86
Number of tRNA genes		37
Number of rRNA genes		8
Duplicate genes		18
GC content	Total (%)	37.2
	LSC (%)	35
	SSC (%)	31.7
	IR (%)	42.9
	CDS (%)	37.9
	rRNA (%)	55.2
	tRNA (%)	53.2
	ALL gene %	39.4
Protein-coding part (CDS) (% bp)		51.1
All genes (% bp)		71.6
Non-coding region (% bp)		28.4

**Table 2 genes-13-00461-t002:** Genes predicted in the chloroplast genome of *Benincasa hispida*. The number of asterisks after the gene names indicates the number of introns contained in the genes.

Category of Genes	Group of Genes	Gene Name
Photosynthesis-related genes	Large subunit of rubisco	*rbcL*
	Photosystem I	*psaA*, *psaB*, *psaC*, *psaI*, *psaJ*
	Assembly/srability of photosystem I	*ycf3 ***, *ycf4*
	Photosystem II	*psbA*, *psbB*, *psbC*, *psbD*, *psbE*, *psbF*, *psbH*, *psbI*, *psbJ*, *psbK*, *psbL*, *psbM*, *psbN*, *psbT*, *psbZ*
	ATP synthase	*atpA*, *atpB*, *atpE*, *atpF **, *atpH*, *atpI*
	Cytochrome b6/f complex	*petA*, *petB **, *petD **, *petG*, *petL*, *petN*
	Cytochrome c synthesis	*ccsA*
	NADH dehydrogenase	*ndhA **, *ndhB **, *ndhC*, *ndhD*, *ndhE*, *ndhF*, *ndhG*, *ndhH*, *ndhI*, *ndhJ*, *ndhK*
Transcription and translation related genes	RNA polymerase subunits/transcription	*rpoA*, *rpoB*, *rpoC1 **, *rpoC2*
	Small subunit of ribosomal proteins	*rps11*, *rps12 * (*2)*, *rps14*, *rps15*, *rps16 **, *rps18*, *rps19*, *rps2*, *rps3*, *rps4*, *rps7* (*2), *rps8*
	Large subunit of ribosomal proteins	*rpl14*, *rpl16 **, *rpl2 * (*2)*, *rpl20*, *rpl22*, *rpl23* (*2), *rpl32*, *rpl33*, *rpl36*
	Translation initiation factor	*infA*
RNA genes	Ribosomal RNA	*rrn16* (*2), *rrn23* (*2), *rrn4.5* (*2), *rrn5* (*2)
	transfer RNA	*trnA-UGC * (*2)*, *trnR-ACG (*2)*, *trnR-UCU*, *trnN-GUU (*2)*, *trnD-GUC*, *trnC-GCA*, *trnQ-UUG*, *trnE-UUC*, *trnG-GCC*, *trnG-UCC **, *trnH-GUG*, *trnI-CAU (*2)*, *trnI-GAU * (*2)*, *trnL-CAA (*2)*, *trnL-UAA **, *trnL-UAG*, *trnK-UUU **, *trnfM-CAU*, *trnM-CAU*, *trnF-GAA*, *trnP-UGG*, *trnS-GCU*, *trnS-GGA*, *trnS-UGA*, *trnT-GGU*, *trnT-UGU*, *trnW-CCA*, *trnY-GUA*, *trnV-GAC (*2)*, *trnV-UAC **
Other genes	RNA processing	*matK*
	Carbon metabolism	*cemA*
	Fatty acid synthesis	*accD*
	Proteolysis	*clpP1 ***
	Component of TIC complex	*ycf1 (*2)*
	Hypothetical proteins	*ycf2 (*2)*

* Gene with one intron, ** gene with two introns, (*2) gene with two copies.

**Table 3 genes-13-00461-t003:** Likelihood ratio tests of five potential genes under positive selection.

Gene Name	Models	np	ln L	Likelihood RatioTest *p*-Value	Positively Selected Sites	
					AA-Site	Score
*accD*	M8 (beta)	10	−2173.400149	0.007931755	159 W	0.984 *
	M7 (beta & ω > 1)	8	−2178.23703			
*clpP*	M8 (beta)	10	−926.578492	0.070969008		
	M7 (beta & ω > 1)	8	−929.224004			
*rps4*	M8 (beta)	10	−877.349259	0.030217199	158 Q	0.971 *
	M7 (beta & ω > 1)	8	−880.848603			
*ycf1*	M8 (beta)	10	−6139.981658	0.156641895		
	M7 (beta & ω > 1)	8	−6141.835451			
*ycf2*	M8 (beta)	10	−9230.970637	0.063743376		
	M7 (beta & ω > 1)	8	−9233.723527			

np represents degree of freedom; ln L represents log likelihood values; *: empirical Bayes values > 0.95.

## Data Availability

The data that support the findings of this study are openly available in the GenBank of NCBI at https://www.ncbi.nlm.nih.gov (accessed on 13 January 2022), reference number (MW362306).

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
