# Peer review of "Comparative Chloroplast Genome Analysis of Wax Gourd (Benincasa hispida) with Three Benincaseae Species, Revealing Evolutionary Dynamic Patterns and Phylogenetic Implications"

_genes, 2022, doi:10.3390/genes13030461_

Round 1
Reviewer 1 Report
After numerous corrections made by the Authors, the consistency of the article and the quality of the presentation of the results have improved significantly. I consider the manuscript correct。
Reviewer 2 Report
A detailed characteristics, the structural variation, and phylogenetic position of wax gourd (Benincasa hispida) among related species of Benincaseae was given by the authors based on molecular markers of chloroplast genome of B. hispida.
Correction of the manuscript improved even further a reliable foundation for phylogenetic studies of Cucurbitaceae family and clarified the taxonomical discrepancies as well as phylogenetics positions among related species.
I confirm my evaluation from my previous report and I recommend the manuscript to be published in Genes.
This manuscript is a resubmission of an earlier submission. The following is a list of the peer review reports and author responses from that submission.
Round 1
Reviewer 1 Report
The revision of the manuscript entitled: “Comparative chloroplast genome analysis of wax gourd (Benincasa hispida) with three Benincaseae species, revealing evolutionary dynamic patterns and phylogenetic implications”
The scientific problem undertaken by the Authors in the manuscript is a justified subject of research, as our knowledge of the characteristics and variability of plastomes still requires a huge amount of work and with each taxonomic group that has not been studied in this respect so far, a new field of research for researchers opens up. I believe that it is a very good idea to describe the new genome together with its comparative analysis against known genomes of closely related species. In my opinion the strong point of the manuscript is the analysis of the presence of microsatellites in the studied plastomes, as well as the visualization of genome divergence in a sequence identity plot. Despite the general right concept of the study, in the manuscript I came across aspects that lower its scientific value and require improvement. Below I point out some individual remarks and suggestion, which could contribute to improving the quality of the manuscript.
- I recommend an in-depth linguistic proofreading.
- The Introduction describes the subject of research: the studied species and the analyzed plastid genome itself, but does not introduce the reader to the problems of molecular research on Cucurbitaceae or Benincaseae. Apart from the will to learn about the variability, no research hypothesis has been formulated. The reader does not know if there are any problems with the molecular or morphological identification of taxa, what is the state of research on the phylogenesis of this group, are there any predictions about the selection pressure?
- Line 66-67. Please, rewrite the sentence, because in present form it suggests that gene content is dependent on nucleotide substitutions.
- Line 69. The cp genome does not “perform” anything. A stylistic correction is needed here.
- Line 70-71. “Authentic, time-effective, cost-effective” – compared to what? Please, precise what You had on Your mind.
- Line 147: “Slicing Window” – are You sure it shouldn’t be sliding window?
- Line 209: When You use CDS, the word “genes” is redundant here.
- Line 214: Fifty-four and fifty-six?
- Line 266: IGS – Please, reconsider if “Intra gene sequences” is an appropriate expansion of IGS shortcut here. I would use “Intergenic sequence” instead.
- The analysis of nonsynonymous versus synonymous substitutions ratio is a very simplified test and susceptible to misinterpretation, especially with longer genes. To conclude about the type of selection pressure, hypothesis testing is commonly performed by comparing the observed Ka/Ks ratio to null models. I recommend EasyCodeML for this purpose, or at least the Data Monkey platform.
- Line 318. “The number of accD gene” – I gues “the result obtained” or “the value” would fit better here.
- Line 338: Cucumis is also a sister group for Benincasa in presented phylogenetic trees.
- Line 397-398: “The non-coding regions were generally agreed to be more conserved than coding regions” – I guess it is a simple but striking mistake.
- Line 411: Please, explain, what do You mean by “slightly phylogenetic order”?
- In my opinion The Discussion chapter would benefit much by citing more concrete examples of particular gene variability on a tribe level. There are no specific examples of comparing the level of variability, e.g. ycf1, mtK or accD. Also the subject of amino-acid composition and potential selective pressure detected in some genes needs a thorough discussion which practically is missing here.
The very idea of the research and the planned methodology can be the basis for a correct publication, and it need make the necessary corrections and improvements.
Author Response
Dear reviewer,
Thank you for your recognition and guidance of our work. We highly appreciate your opinions and have made corresponding corrections and improvements for each point.
Point 1: linguistic problems:
1.I recommend an in-depth linguistic proofreading.
3.Line 66-67. Please, rewrite the sentence, because in present form it suggests that gene content is dependent on nucleotide substitutions.
4.Line 69. The cp genome does not “perform” anything. A stylistic correction is needed here.
5.Line 70-71. “Authentic, time-effective, cost-effective” – compared to what? Please, precise what You had on Your mind.
6.Line 147: “Slicing Window” – are You sure it shouldn’t be sliding window?
7.Line 209: When You use CDS, the word “genes” is redundant here.
8.Line 214: Fifty-four and fifty-six?
9.Line 266: IGS – Please, reconsider if “Intra gene sequences” is an appropriate expansion of IGS shortcut here. I would use “Intergenic sequence” instead.
11.Line 318. “The number of accD gene” – I gues “the result obtained” or “the value” would fit better here.
12.Line 338: Cucumis is also a sister group for Benincasa in presented phylogenetic trees.
13.Line 397-398: “The non-coding regions were generally agreed to be more conserved than coding regions” – I guess it is a simple but striking mistake.
14.Line 411: Please, explain, what do You mean by “slightly phylogenetic order”?
Response 1:
We highly appreciate you pointing out our linguistic shortcomings with some detailed suggestions for improvement.
For each point you raised, we revised the manuscript accordingly in response to each comment. Errors and unclear semantics have been revised. In addition to these problems, we also double-checked the entire manuscript in order to make the language of the article as clear and understandable as possible.
Point 2: The Introduction describes the subject of research: the studied species and the analyzed plastid genome itself, but does not introduce the reader to the problems of molecular research on Cucurbitaceae or Benincaseae. Apart from the will to learn about the variability, no research hypothesis has been formulated. The reader does not know if there are any problems with the molecular or morphological identification of taxa, what is the state of research on the phylogenesis of this group, are there any predictions about the selection pressure?
Response 2:
Based on your comments, we have added molecular research on Cucurbitaceae to the Introduction.
The early molecular phylogeny of Cucurbitaceae was reconstructed using five chloroplast markers, which weakly support two subfamilies of Cucurbitoideae and Nhandiroboideae [4]. Recent studies have reported that the phylogenetic tree of Cucurbitaceae contains a new classification of 15 tribes and 95-97 genera, using 14 molecular markers from the nuclear, plastid and mitochondrial genomes [5]. However, the backbone relationships of tribes are still unresolved possibly due to limited phylogenetic signals of the molecular markers with a large proportion (over 70%) of missing data [6]. Comprehensive and complete sequence information is a reliable foundation for phylogenetic studies of Cucurbitaceae.
[4].Kocyan, A.; Zhang, L.B.; Schaefer, H.; Renner, S.S. A multi-locus chloroplast phylogeny for the Cucurbitaceae and its implications for character evolution and classification. Mol. Phylogenet. Evol. 2007, 44,553-577.
[5].Renner, S.S.; Schaefer, H. Phylogeny and Evolution of the Cucurbitaceae. In: Grumet R., Katzir N., Garcia-Mas J. (eds) Genetics and Genomics of Cucurbitaceae. Plant Genetics and Genomics: Crops and Models. 2016, 20, 155–172.
[6].Guo, J.; Xu, W.; Hu, Y.; Huang, J.; Zhao, Y.; Zhang, L.; Huang, C.H.; Ma, H.; Phylotranscriptomics in Cucurbitaceae reveal multiple whole-genome duplications and key morphological and molecular innovations. Mol. Plant. 2020, 13, 1117–1133.
Point 3: Selection pressure section:
- The analysis of nonsynonymous versus synonymous substitutions ratio is a very simplified test and susceptible to misinterpretation, especially with longer genes. To conclude about the type of selection pressure, hypothesis testing is commonly performed by comparing the observed Ka/Ks ratio to null models. I recommend EasyCodeML for this purpose, or at least the Data Monkey platform.
- In my opinion The Discussion chapter would benefit much by citing more concrete examples of particular gene variability on a tribe level. There are no specific examples of comparing the level of variability, e.g. ycf1, mtK or accD. Also the subject of amino-acid composition and potential selective pressure detected in some genes needs a thorough discussion which practically is missing here.
Response 3:
This perspective is very helpful.Because it refines the approach to our inquiry and makes our results discussion section more informative. We are very grateful to you for your highly valuable suggestions. In response, we used EasyCodeML to perform a selection pressure analysis on our data. In particular, as you mentioned, accD gene was indeed an subject to more positive selection pressure. We also further discussed this point in the article and cited the some corresponding articles.
Finally, we would like to again emphasize our appreciation for your deep insight and guidance, which certainly makes our article more valid in the field of plant chloroplast genomes.
Best regards,
Chao Shi.

Reviewer 2 Report
Manuscript titled: “Comparative chloroplast genome analysis of wax gourd (Benincasa hispida) with three Benincaseae species, revealing evolutionary dynamic patterns and phylogenetic implication” by Weicai Song and coworkers, submitted to Genes, gives results on the detailed characteristics, the structural variation, and on the evolution of the chloroplast genome of three Benincaseae species.
In the Introduction the authors show the importance of Benincasa hispida - wax gourd and related species, as widely distributed in many subtropical and tropical countries. The wax gourd with tasty, edible fruits used like zucchini has not only nutritional but also medical applications. Moreover, the Introduction gives, in a concise way, a general description of the structure and characteristics of chloroplast (cp) genome, cp genes and their utility for evolutionary studies. Therefore, analysis of taxonomy and phylogenetic position among related species of Benincaseae is an important subject of investigations.
Methods are correctly applied and allow to perform a detailed comparative analysis of cp genomes in Benincaseae.
As main findings I consider elaboration of the complete structure and the composition of different regions and their functions in B. hispida. I highly evaluate comparison of similarities and divergences in genome structures, putative RNA editing sites; patterns of microsatellite and oligonucleotide repeats in the chloroplast genomes among Benincaseae. I appreciate phylogenetic relationships among Cucurbitaceae within Benincaseae established in the manuscript. In particular, the phylogenetic analysis, based on complete cp genomes, suggested the sister relationships of B. hispida with few other genera. The possibility to correct the taxonomic discrepancies within the phylogenetics among species of Cucurbitaceae is, in my opinion, the most important feature of this work.
Concluding:
I highly evaluate the results obtained in the manuscript titled: “Comparative chloroplast genome analysis of wax gourd (Benincasa hispida) with three Benincaseae species, revealing evolutionary dynamic patterns and phylogenetic implication” by Weicai Song and coworkers. A substantial step towards the evolutionary characteristics of chloroplast genome structure of important economic fruit crop in Asia and several tropical countries was made which helped to understand phylogenetic patterns within Benincaseae.
Author Response
Thank you very much for your guidance and recognition of our work
It gives us the direction and motivation to continue our research
We really appreciate the time you spared for our manuscript.
Wishing you a wonderful Christmas!
Reviewer 3 Report
Major issues:
- The primary weakness of the paper is that it does not provide any significant new contributions to understanding of phylogenetic relationships in Benincaseae or Cucurbitaceae, nor to understanding of chloroplast plastid evolution.
- The only data generated in this study was the plastid genome sequence of Benincasa hispida, of which there is already another released plastid genome (MT379887), which was not mentioned in the manuscript.
- Too many wrong words, typos and grammar mistakes, even in abstracts. e.g.
- L19: conversation -> conversion
- L22: variety -> varied
- L28: provide -> provides
- L44: long shortage life? "long storage"?
- L49: remove "places that"
Other issues:
- L43: what do the authors mean by "local markets" when no places were mentioned.
- L112-113: NOVOPlsty -> NOVOPlasty, and it is not a "splicing software". Why not directly use a stand-alone reproducible assembler, but instead used a poor non-reproducible manual method then calibrate the result using a stand-alone software?
- L117: what is the artificial way?
- L157: why is both IR regions included since they are identical and will be overrepresented in the likelihood calculation? This is not typical.
- L161: model comparison were determined based on AIC/AICc/BIC. The BIC is Bayesian information criterion, not the "BI".
Author Response
Thank you for your insights into chloroplast genome researches and we highly respect your judgement regarding our work, which gives us a more profound perspective to re-examine our work.
Revisions regarding your insights has been made and point-to-point details elaborated below:
Major issues:
Our work further confirmed the consistancy and discussed variations of the taxonomy position. The difference of two types of phylogenetic tree construction was also discussed. (According to your review, We also further refined our manuscript regarding this section.
Moreover, we provided the elaborated details of chloroplast genome of Benincasa hispida, and the comparative analysis revealing common patterns of genetic consistency and variations. These results are expected to provide resources and references for future research involving genomics, taxonomy and Cucurbitaceae related studies, etc.
This data you mentioned had not published by NCBI when we were doing our research, so its just a concurrent study.
More importantly, only the sequence was published to the public database by the other team, and no analysis was done. However, we focused on the analysis of genomic characteristics and comparison with other closely related species.
We revised the manuscript accordingly in response to each comment, with a thorough grammatical check and modification of the entire manuscript.
Other issues:
Thank you for bringing to our attention the semantic ambiguity. We have changed the original text “local markets” to “markets”.
The original “splicing software NOVOplsty 4.2” has been modified to “splicing script NOVOplasty 4.2”.
With respect to your consideration, you may also understand that, by far, no splicing method is perfect. In this article, two assembly methods are used just to verify each other to get the optimal solution. This is also the dominant assembly method in the current chloroplast and mitochondria articles.
Sorry for the misunderstanding. That was meant to express “manually checked”. Thanks for reminding us this problem. Corresponding changes have been made in the manuscript.
Because most of the views believe that not only the duplicated IR, but also the highly variable IGS regions cause exaggerated phylogenetic distances. So we hesitated on the choice of tree building methods and finally decided to construct both phylogenetic trees and compare them.
So we performed two types of phylogenetic tree construction based on the complete cp genome and unique gene sequences. Then a comparative analysis was performed for both phylogenetic relationships and their corresponding bootstrap values.
The similarities and differences between the two tree building methods were discussed so that a more plausible phylogenetic relationship can be explored.
Thank you for pointing out mistakes. We appreciate your help.In addition to this mistake, we double-checked the entire manuscript as well to make sure the correctness of each detail.
Thank you again for your insights and help. We wish you a wonderful Christmas!

Round 2
Reviewer 1 Report
Dear Authors,
after carefully reading the revised version of the manuscript, I can say that You responded satisfactorily to my suggestions. The linguistic revision, corrections and additions made, and the extension of the analyzes to the method proposed by me significantly increased the quality of the manuscript, its coherence and scientific soundness. Therefore, I am convinced that the work is suitable for publication as it stands.
I choose minor revision to allow You to correct some minor linguistic mistakes, which are listed below.
Line 30: regarDING
Line 91: HAS been published
Line 358: sliDing window.
Reviewer 3 Report
Simply adding two more phylogenetic citations in the intro and two more sentences in the discussion did not mean adding a good contribution to the understanding of phylogenetic relationships in Benincaseae or Cucurbitaceae, nor answered my concern. The newly added phylogenetic works are true updated ones, but not took into discussion.
The study only generated one plastid genome that seems quite normal in the plastid structure. The plastid genome of exactly the sample species, which was released one year ago, worth a mention, if no authorization from the author of MT379887.
There are still obvious typos and grammar mistakes, even in abstracts. e.g.
L31: provide/resource
L61: firmly agreement
L100: pattens
and more..
Please also carefully check captions and the newly added sentences.
I would not trust the only sequence generated in this study until the authors correctly describe what Novoplasty is.
The duplicated IRs, which has significantly slower subs per cite, will never exaggerate phylogenetic distances, but shrink them.